# Beyond Human Detection: A Benchmark for Detecting Common Human Posture

**DOI:** 10.3390/s23198061

**Published:** 2023-09-24

**Authors:** Yongxin Li, You Wu, Xiaoting Chen, Han Chen, Depeng Kong, Haihua Tang, Shuiwang Li

**Affiliations:** Guangxi Key Laboratory of Embedded Technology and Intelligent Information Processing, College of Information Science and Engineering, Guilin University of Technology, Guilin 541006, China

**Keywords:** human detection, common human posture detection, CHP dataset, benchmark

## Abstract

Human detection is the task of locating all instances of human beings present in an image, which has a wide range of applications across various fields, including search and rescue, surveillance, and autonomous driving. The rapid advancement of computer vision and deep learning technologies has brought significant improvements in human detection. However, for more advanced applications like healthcare, human–computer interaction, and scene understanding, it is crucial to obtain information beyond just the localization of humans. These applications require a deeper understanding of human behavior and state to enable effective and safe interactions with humans and the environment. This study presents a comprehensive benchmark, the Common Human Postures (CHP) dataset, aimed at promoting a more informative and more encouraging task beyond mere human detection. The benchmark dataset comprises a diverse collection of images, featuring individuals in different environments, clothing, and occlusions, performing a wide range of postures and activities. The benchmark aims to enhance research in this challenging task by designing novel and precise methods specifically for it. The CHP dataset consists of 5250 human images collected from different scenes, annotated with bounding boxes for seven common human poses. Using this well-annotated dataset, we have developed two baseline detectors, namely CHP-YOLOF and CHP-YOLOX, building upon two identity-preserved human posture detectors: IPH-YOLOF and IPH-YOLOX. We evaluate the performance of these baseline detectors through extensive experiments. The results demonstrate that these baseline detectors effectively detect human postures on the CHP dataset. By releasing the CHP dataset, we aim to facilitate further research on human pose estimation and to attract more researchers to focus on this challenging task.

## 1. Introduction

Human detection is an important task in computer vision, focusing on localizing and identifying humans within given images or videos [1,2]. It serves as the foundation for numerous downstream computer vision tasks, such as image captioning, robotics, and human–machine interaction, among others. The primary goal of human detection is to create computational models and techniques that offer essential location information for computer vision applications. Thanks to the implementation of deep neural networks, significant progress has been made in the field of human detection in recent years [3,4,5].

However, simply possessing human localization information is inadequate for more sophisticated computer vision tasks. For example, scene understanding necessitates the recognition of an individual’s pose, state, and action in order to incorporate significant information at various levels and to derive semantic connections, roles, and events within the scene [6,7]. The goal of autonomous driving is to develop and build a vehicle capable of navigating independently without driver supervision. The ability to detect potentially hazardous situations and anomalies is vital for driving safety and demands more information than just human localization [8,9]. Service robots interact with humans in various ways, often aiming to enhance the consumer experience by providing assistance, providing guidance, or performing specific tasks [10,11].

The interaction between the service receiver and the robot is a determining factor in the user’s emotional experience with the service provider. To optimize the use of service robots, it is crucial for these robots to obtain more information than just the localization of the humans they serve. By acquiring additional data, such as user sitting and standing postures, gesture, action, and emotions, service robots can better understand and respond to individual needs, ultimately enhancing user experience and satisfaction [12]. Elderly guardianship by robots has the potential to revolutionize elderly care by providing personalized, round-the-clock assistance and support, which refers to the use of robotic technology and artificial intelligence to assist, support, and care for older adults, particularly those with limited physical or cognitive abilities [13,14] These robots, often referred to as “carebots” or “elder care robots”, can provide various services to improve the quality of life for seniors and to support their independence while ensuring their safety and well-being [15,16]. They are expected to provide physical care such as assistance with tasks like bathing, dressing, and meal preparation; medical care such as monitoring the elderly person’s health and ensuring that they receive appropriate medical attention; and emotional support such as providing companionship, social interaction, and mental stimulation to help combat loneliness and depression [17,18]. It is, therefore, essential for these robots to obtain information beyond the localization of elders in these applications.

To draw attention to human detection beyond localization for more advanced computer vision tasks, in this paper, we are concerned with the posture as well as localization of a human being, which is crucial for many applications, as just mentioned above. Human posture or pose estimation has a variety of advanced computer vision applications, including intelligent robot services [19,20,21], intelligent sports coaches [22,23,24,25], intelligent security monitoring systems [26,27,28], virtual reality [29,30,31], and intelligent medical rehabilitation [30,32,33]. Its purpose is to locate the joints of the human in images and videos, that is, to search for specific poses within the space of all joint poses. However, accurate pose estimation is difficult for images lacking texture information. In this paper, we propose and motivate a task that lies between human detection and human pose detection, which applies to a wider range of advanced computer vision applications and provides more information as an upstream task for other tasks.

Due to the remarkable advancements in deep learning and the availability of extensive training data, great progress has been made in human detection and localization technologies in recent years [2,3,4,5]. Unfortunately, human detection can only offer the fundamental elements of information that computer vision applications necessitate, such as the locations of humans within a scene. However, this information alone is inadequate for more intricate computer vision tasks. For example, recognizing the posture of person is crucial to extracting high-level semantics for the task of scene understanding [34]. Estimating the pose of persons underpins a variety of applications of human activity estimation, robotics, motion tracking, augmented reality, etc. [35,36,37,38]. Human pose estimation generally uses the keypoint estimation method to select a set of most representative points in human pose, such as the head, shoulders, elbows, wrists, hips, knees, ankles, etc., and portrays the human pose by connecting the lines [39,40]. However, the accurate identification of these joints, especially through manual methods, greatly depends on the presence of comprehensive texture data. It is very difficult to manually identify the joints of human bodies when the subject is wearing loose or bulky clothing, which can obscure the joints and make them difficult to locate. Additionally, overlapping body parts, poor lighting conditions, and complex backgrounds can make it harder to discern the exact position of joints. In cases where the subject is engaged in rapid or complex movements, it may also be challenging to track joint positions accurately over time. Without sufficient well-annotated data, developing and evaluating big deep learning models are hardly possible.

In order to address the aforementioned challenges, we formulate a new task that is a compromise between human detection and human pose estimation. This task aims to provide additional information beyond the mere location of detected individuals and to avoid difficulties in collecting well-annotated data for human pose estimation. Specifically, this paper focuses on the localization of humans and the recognition of their postures in RGB images. These postures are divided into seven common categories, i.e., bending, squatting, sitting, running, going, standing, and lying. Figure 1 shows the difference between traditional human detection and our common human posture detection. As the posture of individuals being provided, this task finds potential applications in diverse areas, including gaming [41], AI-powered personal trainers [42,43], robotics [38,44], video surveillance [45,46], etc. We believe that this task holds significant potential for advancing computer vision perception, analysis, and interpretation, potentially inspiring further investigation into novel detection tasks beyond identification and localization. This task is referred to as common human posture detection (CHPD). To facilitate future research on this task, we introduce a dataset called CHPD, which contains seven types of human posture of bounding-box annotations collected from 5250 RGB images. With this well-annotated dataset, we have developed two baseline detectors, namely CHP-YOLOF and CHP-YOLOX, building upon two identity-preserved human posture detectors, IPH-YOLOF and IPH-YOLOX [47], and conduct comparative ablation experiments to evaluate their performance.

The key contributions of our work can be outlined as follows:We formulate a novel task of common human posture detection, which underpins a variety of applications where information about human posture is desired but traditional pose estimation is hard and which may also attract attention to more informative object detection methods that extend beyond mere identification and localization.We introduced CHP dataset, the first benchmark dedicated to common human posture detection.We developed two baseline detectors, i.e., CHP-YOLOF and CHP-YOLOX, based on two identity-preserved human posture detectors to support and stimulate further research on CHP.

## 2. Related Work

### 2.1. Traditional Human Detection

In traditional human detection methods, many researchers have been enthusiastic about utilizing manually designed human features and classifiers for human detection and localization. For instance, Dalal et al. [48] proposed the Histogram of Oriented Gradients (HOG) algorithm, which extracts human edge and texture features by computing the histograms of local gradient orientations in the image. Wang et al. [49] introduced a local image descriptor called LPP-HOG (Locality Preserving Projection-HOG) for fast human detection. This method applies Locality Preserving Projection to the HOG feature vector to obtain a low-dimensional LPP-HOG vector, which is then used as input to a linear SVM classifier. Shen et al. [50] proposed an enhanced variable-size HOG feature based on boosting human detection models. The proposed feature leverages the information that is overlooked in the quantization of gradient orientations and integrates it with a fixed Gaussian template and integral orientation histograms. Wang et al. [51] combined Histogram of Oriented Gradients (HOG) and Local Binary Patterns (LBP) as feature sets and proposed a human detection model capable of handling partial occlusion. The model employs a linear Support Vector Machine (SVM) to learn global detectors for the entire scanning window and partial detectors for local regions from the feature sets. Pang et al. [52] presented two methods to accelerate the HOG algorithm for human detection. One method reuses features within blocks to construct HOG features for overlapping detection windows, and the other method effectively computes HOG features for each block using sub-unit based interpolation. Ye et al. [53] proposed a Piecewise Linear Support Vector Machine (PL-SVM) approach, which constructs a linear classification boundary using piecewise decision functions. Each piecewise SVM model is responsible for a subspace corresponding to a specific viewpoint or pose of humans. This method discriminates multiple viewpoints and poses of humans from the background in a high-dimensional feature space.

Although traditional human detection algorithms have some application value in simple scenarios, with the development of deep learning techniques, algorithms based on deep neural networks have achieved better performance, demonstrating their capability to handle complex scenes and pose variations.

### 2.2. Deep Learning Methods for Human Detection

Due to the powerful feature representation and adaptability of deep learning detection algorithms, they can handle challenges such as illumination variations and pose changes in complex scenes, exhibiting excellent generalization capabilities. As a result, a plethora of deep learning network models have been developed, including the one-stage You Only Look Once (YOLO) series [54,55,56,57,58,59] and the two-stage Region-Convolutional Neural Network (R-CNN) series [60,61,62]. Numerous researchers have devoted themselves to applying deep learning networks to human detection tasks. For instance, Nikouei et al. [63] employed the region-based Fast R-CNN [60] for pedestrian detection. They first extracted image features using CNN and then established a region proposal network to extract potential regions containing pedestrians, combined with K-means clustering analysis. Zhao et al. [64] developed a two-stage method that utilizes depth data for real-time human detection. This approach initially utilizes their proposed Physical Radius Depth (PRD) detector for rapid detection of human candidates, followed by applying a CNN to extract human features and subsequently refine the genuine human candidates based on the CNN features. Lan et al. [65] improved the YOLO [54] algorithm and introduced a new network structure, YOLO-R. They incorporated three Passthrough layers composed of Route and Reorg layers into the original YOLO network to connect shallow and deep layers, as well as high-resolution and low-resolution pedestrian features, reducing the loss of human features. In [66], the researchers utilized an adaptive YOLO network to improve the performance of detecting handball players. They evaluated multiple YOLO-based models and network training configurations with different datasets to improve the detection results of player identification in handball images. In order to address incomplete human detection in specific video frames, Zhou et al. [67] proposed an enhanced YOLO v4 algorithm, which incorporates the Ghost module in the CBM module to reduce the number of parameters further.

In summary, in the domain of object detection using a single neural network, the YOLO series dominates. YOLO treats object detection as a regression task, predicting bounding boxes and corresponding class probabilities. Compared with the R-CNN series, YOLO excels in terms of real-time performance and simplicity. Hence, in this study, we have developed two baseline detectors, namely CHP-YOLOF and CHP-YOLOX, by utilizing two identity-preserved human posture detectors: IPH-YOLOF and IPH-YOLOX. These detectors inherit their network architecture from the two most common state-of-the-art YOLO variants.

### 2.3. Human Pose Estimation

Traditional methods with human pose estimation mainly rely on handcrafted feature extraction and machine learning algorithms such as random forests and Support Vector Machines (SVM). However, these methods are limited by the representational and generalization capabilities of handcrafted features, imposing constraints on complex scenes and posing variations. In recent years, deep learning techniques have made significant breakthroughs in human pose estimation. Deep learning-based approaches leverage network structures such as convolutional neural network (CNN) or recurrent neural network (RNN) to learn mapping relationships from images to end-to-end poses. Some notable deep learning-based human pose estimation methods include OpenPose [68], Stacked Hourglass Network [69], EfficientPose [70], and others. Currently, most human pose estimation algorithms focus on predicting the coordinates of human keypoints, i.e., keypoint localization, to characterize human poses by determining spatial relationships between keypoints based on prior knowledge. For example, Wei et al. [71] proposed the Convolutional Pose Machines (CPM) method, which optimizes keypoint position estimation progressively through cascaded convolutional networks. Sun et al. [72] proposed an integral regression approach for human pose estimation, enhancing the accuracy by predicting keypoint offsets at multiple scales. However, recognizing human keypoints, especially manually identifying them, relies on rich texture information. The influence of different environments (e.g., lighting, occlusion) and camera angles leads to missing features of human instances in images, making it challenging to identify keypoints of human instances in the images. Developing and evaluating large-scale deep learning models for human pose estimation are impossible without sufficient and diverse annotated data. In sum, it is difficult to use keypoint-based methods to estimate pose for images with complex backgrounds or missing features. In view of this, in this paper, we propose a new task between human body detection and human pose estimation, i.e., common human posture detection.

## 3. Detection Benchmark for Human Posture

Our objective is to create a specialized dataset specifically designed for the identification of common human postures (CHP). In the process of developing CHP, we ensure the inclusion of a wide variety of scenarios and furnish each image with manual annotations, as described subsequently. The CHP dataset is available at https://github.com/wuyou3474/CHPDataset (accessed on 27 August 2023).

### 3.1. Image Collection

Acquiring a large and diverse dataset of images is essential for effectively training and evaluating machine learning models for detecting common human postures. To facilitate the process, we have gathered all the necessary images from the CHP dataset available on the Internet, which adequately fulfills our requirements, we follow the principles of the PASCAL VOC [73] and COCO [74] datasets in collecting our dataset CHP. Because iconic images may lack important contextual information and non-normative viewpoints, we cleaned and processed the collected images to remove images with non-normative viewpoints. Human postures are classified into bending, standing, sitting, lying, squatting, running, and going, considering their prevalence, usefulness for practical applications, and ease of image collection. See Figure 2 for sample images of our dataset. Figure 3 shows images that are not suitable for this type of task, all of which are broadly representative. It is evident that while all of these images have distinct human features, the posture features are either overlapping, or severely missing, or blurred. For example, both images (1) and (2) in Figure 3 suffer from overlapping posture features. Images (3) through (10) are characterized by missing posture features. Although the legs are out-of-view in image (5), we can infer that the person is running. The images (11) and (12) have the problem of blurred and missing features. In general, we filter out the images that we are unable to infer the posture of the persons without ambiguity.

### 3.2. Annotation

In this section, we will describe how to annotate the collected images. Based on the proposed task of detecting common human postures, the image annotations necessitate the subsequent attributes:**Category:** human.**Common Human Posture box:** one of ‘bending’, ‘lying’, ‘going’, ‘running’, ‘sitting’, ‘squatting’, and ‘standing’.**Bounding box:** an axis-aligned bounding box surrounding the extent of the human visible in the image

Following the annotation guidelines proposed by PASCAL VOC and the annotation rules of COCO, we divided the annotations into three steps: manual annotation, inspection, and correction. Manual annotation was accomplished through the collaborative efforts of our annotation team. Owing to potential inaccuracies and mistakes in each member’s annotations, our verification team examined if the annotated information for each image adhered to the guidelines. In case any images failed to meet the specifications, the correction team undertook the necessary adjustments to the image. Figure 4 illustrates our annotation process. The manual annotation team classified and annotated the images, and handed over the annotated image set to the validation team. If there were category errors or omissions in labeling, the image was recorded and submitted to the correction team for correction. Our CHP dataset ensures a high quality of annotations through the above steps. Please refer to Figure 2 for an example of the annotation for CHP.

### 3.3. Dataset Statistics

The CHP dataset contains a diverse range of scenes, including both public and private spaces, as well as variations in distance, perspective, lighting conditions, etc. The dataset includes 5250 human images, which are divided into two main subsets: a train set and a test set, with an 8:2 ratio (4200 images for training and 1050 images for testing) to facilitate training and performance evaluation. Figure 5a illustrates the distribution of different poses in the training and test sets of the CHP dataset, while Figure 5b depicts the average number of each human pose per image in both the dataset and the test set. Due to uneven data collection, the number of images for each pose varies. Although we were concerned about potential sample imbalance, our experimental results indicate that this distribution reflects the actual distribution of human poses in everyday life, alleviating our concerns. The CHP dataset differs from traditional datasets in that standard annotation boxes only annotate a single object or object feature, whereas our CHP annotations encompass two pieces of information: humans and their poses.

## 4. Baseline Detectors for Detecting Common Human Postures

We propose two baseline detectors to promote the development of detecting common human postures, whose network architecture was inherited from IPH-YOLOF and IPH-YOLOX [47], which were proposed for identity-preserved human posture detection. The researchers in [47] improved two state-of-the-art YOLO variants (YOLOX [59] and YOLOF [58]) by introducing an extra classification head to each original model. This modification enables the models to predict the posture of individuals in thermal images. These two detectors were adapted for our research, and we have improved them in various ways as a result. Specifically, in order to improve the performance of IPH-YOLOX, we use CoaT (Co-Scale Conv-Attentional Image Transformers) [75] or InternImage [76] as the its backbone network to replace its original one, i.e., CSPDarkNet and the Spatial Pyramid Pooling(SPP) [77]. CoaT is built on cross-scale attention and efficient conv-attention operations, and InternImage utilizes deformable convolution as its core operator, allowing its model to possess a large effective receptive field while also having adaptive spatial aggregation based on input and task information. It should be pointed out that the IPH-YOLOX model relies on its ability to fuse multiple-level features by utilizing PANet [78] as the neck sub-network, which is an improved version of FPN. However, IPH-YOLOF only utilizes the highest-level features for detection, limiting spatial aggregation capabilities and the interactions between features of different scales. To improve the performance of the IPH-YOLOF, we introduce the ECA (Efficient Channel Attention) module [79] or Convolutional Block Attention Module with ECA (CBAME) module to its extra head. ECA is based on a local cross-channel interaction strategy without dimensionality reduction. The Convolutional Block Attention Module (CBAM) [80] sequentially infers attention maps along two separate dimensions, channel and spatial, which is a simple yet effective attention module for feed-forward convolutional neural networks. In our proposed CBAME module, we used ECA module as the channel dimension to achieve the better performance. Due to the fact that IPH-YOLOF incorporated with CBAME and IPH-YOLOX incorporated with InternImage achieve better results, as reported in Section 5.3, we use them as the default settings in this paper. Therefore, the proposed baseline detectors are named CHP-YOLOF and CHP-YOLOX, respectively, and their detailed descriptions are provided below.

### 4.1. CHP-YOLOF

Figure 6 displays the network structure of CHP-YOLOF we proposed in this paper. ResNet50 [81], pre-trained on ImageNet [82], is used as the CHP-YOLOF’s backbone network. The backbone network outputs a C5 feature map, which has 2048 channels and a downsampling multiplicity of 32. The dilated encoder of the neck sub-network is responsible for receiving and performing the encoding process on these features. The final decoding module contains two concurrent task-specific heads for classification and regression. Based on the IPH-YOLOF model, an extra classification head is also introduced to predict the common human posture. To achieve a better performance in common human posture estimation, we introduce the CBAME Module in the extra head, which is a novel combination of the ECA and CBAM modules that we proposed. The network architecture of this module is depicted in Figure 7. Specifically, by using 1D convolution, ECA achieves a local cross-channel interaction strategy without dimensionality reduction, which only involves a handful of parameters while bringing clear performance gains from this strategy. The CBAM consists of two consecutive sub-modules, called the channel and spatial sub-modules. At each convolutional block of deep networks, the CBAM module adaptively refines the intermediate feature map. In our proposed CBAME, we use ECA as its channel sub-module to replace origin one (i.e., origin one utilizes both max-pooling outputs and average-pooling outputs with a shared network) for better performance. The loss function for training the CHP-YOLOF model is defined as follows:(1)$$L_{total}=L_{cls}+L_{reg}+\lambda L_{posture}$$

The losses for classification, regression, and common human posture prediction are denoted by $L_{cls}$, $L_{reg}$, and $L_{poture}$, respectively. Additionally, the constant λ is used as the weight coefficient to represent the loss for the prediction head of common human postures. We follow [47,83,84] to provide the definitions of these losses below:(2)$$\begin{array}{c} L_{cls}=\frac{1}{N_{pos}}\sum_{n=0}^{N_{pos}}FL\left(y_{cls}^{n},p_{cls}^{n}⊗p_{obj}^{n}\right),\\ L_{poture}=\frac{1}{N_{pos}}\sum_{n=0}^{N_{pos}}FL\left(y_{poture}^{n}p_{poture}^{n}⊗p_{obj}^{n}\right),\\ L_{reg}=\frac{1}{N_{pos}}\sum_{n=0}^{N_{pos}}\left(smooth_{L1}\left(b_{t}^{n}-b_{p}^{n}\right)\right) \end{array}$$

The ground truth for the classification and human posture is indicated by the variables $y_{cls}$ and $y_{poture}$, and the predictions for the classification, human posture, and boxes (i.e., is there any person in the box) are represented by $p_{cls}^{n}$, $p_{poture}^{n}$, and $p_{obj}^{n}$. The amount of positive anchor is denoted by $N_{pos}$; the scalar product is denoted by ${⊗}_{p}$; and the ground truth bounding box and the prediction bounding box, respectively, are denoted by $b_{t}^{n}$ and $b_{p}^{n}$. The focal loss [85] and the $smooth_{L1}$ loss functions are represented by FL(·) and $smooth_{L1}$, respectively. The focal loss function is frequently employed to solve the challenge of imbalanced distribution between difficult and easy samples in machine learning models. The $smooth_{L1}$ function is widely used in training deep neural networks as it suppresses outliers, which are at a greater distance from the mean. This capability results in less proneness to the gradient exploding problem by controlling the gradient values.

### 4.2. CHP-YOLOX

Figure 8 displays the network structure of CPH-YOLOX we proposed in this paper. To improve the performance of CHP-YOLOX, we replace the original backbone network of CHP-YOLOX, CSPDarkNet, along with the Spatial Pyramid Pooling (SPP) [77], with InternImage [86] as its new backbone network. Specifically, InternImage utilizes deformable convolution as its core operator, which allows the model to effectively scale to large parameter sizes and to acquire more powerful representations from extensive training data. The basic block design of this module refers to ViTs, incorporating more advanced components such as LN [87], feed-forward networks (FFN) [88], and GELU [89]. In order to acquire hierarchical feature maps, it employs convolutional stem and downsampling layers to resize the feature maps to various scales. For more comprehensive information on this module, please refer to [86]. The backbone network outputs the C3, C4, and C5 feature maps, which have 128, 256, and 512 channels with downsampling multipliers of 8, 16, and 32, respectively. The PANet [78] of the neck sub-network is responsible for receiving these features and combining them with shallow features through a bottom-up path and subsequently with deep features via a top-down path. The final decoding module contains two concurrent task-specific heads for classification and regression, in addition to an extra prediction head we also introduce for common human posture prediction, similar to IPH-YOLOX. The total loss function for training the CHP-YOLOX model is defined as follows:
(3)$$L_{total}=L_{cls}+L_{reg}+L_{obj}+\lambda L_{posture}$$

The losses for classification, regression, confidence of boxes, and common human posture prediction are denoted by $L_{cls}$, $L_{reg}$, $L_{obj}$, and $L_{posture}$, respectively. Additionally, the constant λ is used as the weight coefficient to represent the loss for the prediction head of common human postures. We follow [47,83,84] to provide the definitions of these losses below:(4)$$\begin{array}{c} L_{cls}=\frac{-1}{N_{pos}}\sum_{n=1}^{N_{pos}}y_{cls}^{n}ln\left(\sigma \left(p_{cls}^{n}\right)\right),\\ L_{posture}=\frac{-1}{N_{pos}}\sum_{n=1}^{N_{pos}}y_{posture}^{n}ln\left(\sigma \left(p_{posture}^{n}\right)\right),\\ L_{obj}=\frac{-1}{N_{pos}}\sum_{n=1}^{N_{pos}}y_{obj}^{n}ln\left(\sigma \left(p_{obj}^{n}\right)\right),\\ L_{reg}=\frac{1}{N_{pos}}\sum_{n=1}^{N_{pos}}\left(1-IOU\left(b_{t}^{n},b_{p}^{n}\right)\right) \end{array}$$

The ground truth for the classification, common human posture, and the boxes is indicated by the variables $y_{cls}$, $y_{poture}$, and $y_{obj}$, and the predictions for the classification, common human posture, and boxes (i.e., is there any person in the box) are represented by $p_{cls}^{n}$, $p_{poture}^{n}$, and $p_{obj}^{n}$. The amount of positive anchor is denoted by $N_{pos}$, and the ground truth bounding box and the prediction bounding box are denoted by $b_{t}^{n}$, and $b_{p}^{n}$, respectively. The softmax activation and IOU loss functions are represented by σ and IOU(·), respectively. Due to the fact that the intersection over union (IOU) exhibits the characteristic of scale invariance, it is usually utilized to identify both positive and negative samples, as well as to evaluate the distance between the predicted bounding box (bbox) and the ground truth bbox.

## 5. Evaluation

### 5.1. Evaluation Metrics

The performance of the two baseline detectors is evaluated in the experiment through the measurement of their average precision (AP) and mean average precision (mAP). Additionally, we use the Intersection of Union (IOU) metric to evaluate the accuracy of the predicted bounding boxes compared with the ground truth bounding boxes. The precision for a specific object calculated by determining the area under the precision–recall curve in relation to the coordinate axes is referred to as AP, while the algorithm’s performance in evaluating multiple targets or classes is defined as mAP, which is calculated by summing all individual AP values for each target and dividing it by the total number of targets. Recall refers to the true positive rate, which is the proportion of correct positive predictions out of all the actual positives. Precision represents the positive prediction value, which measures the proportion of correct positive predictions out of all the positive predictions made. Recall and precision are defined as follows: Precision=TP/(TP+FP) and Recall=TP/(TP+FN), respectively. The number of true positive detections is referred to as TP, the number of false positive detections is referred to as FP, and the number of false negative detections is referred to as FN. For a more thorough description, please refer to [73].

According to the COCO evaluation [74], we use three metrics, i.e., 0.5, 0.75, and 0.5 to 0.95, as the IoU threshold. Specifically, the AP value is calculated by considering IoU values from 0.5 to 0.95 with an increment of 0.05, which is denoted as AP@[0.50:0.05:0.95]. Additionally, the AP@0.5 denotes the average precision at an IoU threshold of 0.5, whereas the AP@0.75 represents the average precision at an IoU threshold of 0.75. We use the COCO mAP to evaluate the performance of detectors for detecting common human postures. In this paper, the task we propose is common human postures detection, which combines the conventional task of human detection with the task of human pose estimation into a novel task. Our precision metric for this task must consider predicting both the category and the posture simultaneously, unlike the general object detection precision that only predicts the accuracy of the target category. The precision metrics used to predict the human category, common human posture, and their combination are referred to as $AP_{c}$, $AP_{p}$, and $AP_{cp}$, respectively. When we add the prefix ’m’, it indicates the mean average precision, known as mAP.

### 5.2. Evaluation Results

**Comparison with traditional YOLO models.** To demonstrate the superiority of our proposed method, we conducted a comparison between the original traditional YOLO algorithms, namely YOLOv3 [57] and YOLOv5, and the YOLO variant algorithms, namely YOLOF and YOLOX, in terms of $AP_{c}$ metric. As shown in Table 1, YOLOF is essentially the top-performing detector, although it has a slightly lower APc@0.5 compared with YOLOX. Compared with these two variants of YOLO, YOLOv3 and YOLOv5 exhibit relatively lower performance. Specifically, when comparing the highest values of three performance indicators in the YOLO variant models, YOLOv3 exhibited $AP_{c}$s lower than 10.8%, 12.8%, and 10.2%, respectively, while YOLOv5 demonstrated $AP_{c}$s lower than 4.9%, 5.4%, and 4.3%, respectively. Therefore, it makes sense that we developed our baseline detector based on YOLO variant models.

**Overall performance.** We conducted an extensive evaluation of the CHP dataset using the baseline detectors. As shown in Table 2, the evaluation results are reported using the precision metrics $AP_{c}$, $AP_{p}$, and $AP_{cp}$ defined in Section 5.1. The table clearly shows that CHP-YOLOF is essentially the best detector, although it has slightly lower APs@0.5 and $mAP_{p}$ compared with CHP-YOLOX. Moreover, Table 2 also indicates that the average for predicting common human postures is lower than the precision for predicting human categories for all detectors. Specifically, the differences between $mAP_{c}$ and $mAP_{p}$ are all greater than 9.0%. The CHP-YOLOF detector shows the largest difference of 12.0%, which indicates that detecting common human postures is more difficult and challenging than detecting human themselves. Estimating a common human posture may pose more difficulties compared with distinguishing a human from their surroundings, as the former exhibits greater inter-class variation. Despite the challenges involved in this task, we hope that our study will serve as motivation for other researchers to explore the field of common human posture detection.

**Performance on per posture.** To provide more valuable insights into common human posture detection, we conducted an evaluation of the performance of the two baseline detectors on each posture. We use Table 3 to display the $mAP_{cp}$ of the detectors. In general, all detectors exhibit optimal performance among ‘going’, ‘running’, and ‘sitting’, with all the $mAP_{cp}$ above 50%. Compared to them, the $mAP_{cp}$ of ‘bending’, ‘lying’, and ‘squatting’ have relatively low performance, with all the $mAP_{cp}$ being only greater than 40%, and the $mAP_{cp}$ of ‘standing’ is the worst, less than 40%. We can use the following facts to explain these phenomena: (1) ‘bending’, ‘lying’, ‘squatting’, and ‘standing’ face more intra-class variations than ‘going’, ‘running’, and ‘sitting’ due to potentially more background clustering, occlusion, and larger posture variations for the former; (2) the postures of ‘going’, ‘running’, and ‘sitting’ are generally clear and distinct single objects in each image, usually with a standard target size. Specifically, detecting ‘bending’ and ‘standing’ postures pose significant challenges due to various factors. Despite having the largest amount of data, these postures are susceptible to occlusion, multiple targets with non-standard sizes, and intra-class variation. For instance, when the bending angle is small, the detectors often mistake it for the ‘standing’ posture, further complicating the detection process. Regarding the ‘lying’ posture, besides the potential obstruction and the lack of more training data, the challenge of detecting this posture in the image is heightened by the lack of a complete human body image, leading to less than ideal test results. Similarly, the adequacy of training data is not the sole factor influencing squatting posture. In instances where the squatting angle is too slight, the detector may mistakenly identify it as a standing posture, thereby posing challenges to accurate detection. Despite not having the largest training data, the postures of ‘going’, ‘running’, and ‘sitting’ are typically clear and distinct single objects in each image, often with a standard target size. This characteristic makes the detection of these postures relatively simple. We will consider these factors in our future study to enhance the accuracy of common human posture detection.

**Qualitative evaluation.** In Figure 9, we display the qualitative detection results of the 28 samples by using the two baseline detectors. The first two rows display fourteen samples where the two baseline detectors perform exceptionally well, demonstrating accurate prediction of common human postures. In contrast, the last two rows display fourteen samples where the detectors fail to correctly predict the common human postures. As shown in the figure, we use the black and purple bounding boxes to represent the detection results of CHP-YOLOF and CHP-YOLOX, respectively. The letter preceding the vertical line in the box numbers, such as b|0.93, denotes the predicted category of common human posture (b, l, g, r, si, sq, and st represent ‘bending’, ‘lying’, ‘going’, ‘running’, ‘sitting’, ‘squatting’, and ‘standing’ postures, respectively.), while the decimal number following the vertical line represents the predicted score. The excellent performance of the two baseline detectors in the first two rows can be attributed to the absence of any occlusion, minimal background clustering, and the presence of images with standard target sizes. However, accurately predicting the postures can be challenging due to the presence of a background cluster in the last two rows and the variation in human size. We use the samples in the last row to analyze the potential sources of detection errors. The first example illustrates false and missed detections in the ‘bending’ posture, which can be attributed to challenges such as clustering, small objects, and a very small bending angle. The incorrect detections in the second, third, and fifth samples can be attributed to missed detections, primarily caused by factors such as clustering, occlusion, and the presence of small objects. Similarly, in the fourth sample, occlusion resulted in incomplete human body images, causing the detector to confuse the ‘running’ posture with ‘standing’ and ‘sitting’ postures. In the sixth sample, the detector incorrectly identified the ‘squatting’ posture as the ‘sitting’ posture, likely due to their similarity in appearance. In the last sample, the lack of clear texture features, presence of clusters, and ambiguity led to an incorrect prediction of the ‘standing’ posture. The experimental results indicate that in challenging scenarios, it increases the difficulty for the detector to accurately predict common human postures, resulting in detectors that may incorrectly detect the postures.

### 5.3. Ablation Study

**Impact of the proposed components.** The effect of the proposed components on the detector varies. To determine the most effective components for improving the performance of detectors, we conducted training and evaluation using two baseline detectors on the CHP dataset. We use Table 4 to display the mAPs and APs at fixed IoUs (0.5 and 0.75) of two baseline detectors on the CHP dataset with respect to various proposed components. Based on the data presented in the table, it is evident that the incorporation of ECA or CBAME has resulted in an overall improvement in the performance of CHP-YOLOF. More specifically, when ECA is incorporated, LCD-YOLOF’s $mAP_{c}$, $mAP_{p}$, and $mAP_{cp}$ increase by 1.4%, 4.0%, and 2.6%, respectively. On the other hand, when CBAME is incorporated, there are noticeable increases of 1.6%, 4.7%, and 3.2% in CHP-YOLOF’s $mAP_{c}$, $mAP_{p}$, and $mAP_{cp}$, respectively. Simultaneously, it is clear that the incorporation of CoaTnet or InternImage has led to an overall improvement in the performance of CHP-YOLOX. Specifically, when CoaTnet is incorporated, the $mAP_{p}$ and $mAP_{cp}$ of LCD-YOLOF have increased by 1.0% and 1.2%, respectively, except for its $mAP_{c}$ being slightly lower than the original CHP-YOLOX. Similarly, when InternImage is incorporated, there are noticeable increases of 1.1%, 4.1%, and 3.8% in the $mAP_{c}$, $mAP_{p}$, and $mAP_{cp}$ of CHP-YOLOX. Based on the experiment results, it is indicated that incorporating an ECA or CBAME module into the additional classification head has the potential to improve the performance of the CHP-YOLOF detector. Similarly, using CoaTnet or InternImage as the backbone network for the CHP-YOLOX model would also contribute to its performance enhancement. We can use the following facts to illustrate these phenomena: (1) CoaTnet is built on cross-scale attention and efficient conv-attention operations, and InternImage leverages deformable convolution as its core operator, which enables its model to adapt spatial aggregation based on the input and task information; (2) PANet, an enhancement built upon the framework of FPN, also relies on the ability to effectively fuse features from different levels, which is a crucial factor contributing to the success of FPN; (3) ECA only involves a handful of parameters while bringing clear performance gains from local cross-channel interaction strategies without dimensionality reduction; utilizing the ECA module as its channel dimension, CBAME is able to enhance the adaptive feature refinement of CBAM, thereby improving its overall performance.

**Weighting the loss of predicting common human posture.** To comprehend the impact of the weighting coefficient on predicting common human posture, we evaluate CHP-YOLOF on CHP by using a varying weighting coefficient ranging from 0.2 to 2.0 in increments of 0.2, i.e., λ in Equation (Equation 1), to find an optimal weighting coefficient for predicting common human posture. We use Table 5 to display the mAPs and APs at fixed IoUs (0.5 and 0.75) of CHP-YOLOF on CHP with different weighting coefficients. The line chart depicted in Figure 10 presents the results of the mAP metric, which provides us with a more intuitive and effective comparative analysis. Based on the data presented in the table, it is clear that the optimal average precision (AP) for CHP-YOLOF falls within the range of 1.4 to 2.0. However, it is important to note that obtaining the highest AP values simultaneously at a fixed λ is not possible. Specifically, the variation in λ has little impact on $AP_{c}$, as the difference between the highest and lowest values of $AP_{c}$ does not exceed 3.0%. Nevertheless, noticeable fluctuations in both $AP_{p}$ and $AP_{cp}$ can be discerned as λ varies, and it can be inferred that the alterations in $AP_{p}$ and $AP_{cp}$ are essentially synchronized, implying a strong correlation between $AP_{c}$ and $AP_{cp}$. It is noteworthy that the optimal $AP_{p}$ and $AP_{cp}$ are obtained when λ falls within the range of 1.6 to 2.0. Furthermore, the overall maximum values are obtained at λ = 2.0, which is the default setting. In summary, a value of λ = 1.4 is considered an optimal option for $AP_{c}$s, while a value of λ = 2.0 is considered an optimal option for both $AP_{p}$s and $AP_{cp}$s. The experimental results suggest that simultaneously performing both human detection and common human pose estimation tasks as a composite task can be challenging, as these two tasks may contradict each other. Developing more efficient methods to tackle this challenge is crucial, and it will be a primary focus area for our future research.

## 6. Conclusions

This paper formulates a novel task to identify common human postures, which requires detecting humans and estimating their common poses. This task plays a vital role in comprehending human actions, enabling efficient and secure interactions with both humans and the surrounding environment. To address this task, we have developed a comprehensive benchmark dataset called Common Human Postures (CHPs). This dataset is designed to go beyond human detection and to stimulate more informative and promising research directions. In addition, we have established two baseline detectors, namely CHP-YOLOF and CHP-YOLOX. These detectors serve as initial models to provide a foundation for further exploration and development in this challenging task. We anticipate that our work will attract increased attention in the field of identifying common human postures, as this task holds significant implications for advanced applications such as autonomous driving, elderly guardianship systems, and hospital care.

However, it is important to note that the current performance of our proposed network architectures is still not satisfactory. In future work, our aim is to investigate better techniques to reduce the conflict between localizing humans and estimating their postures, with the ultimate goal of improving the detection accuracy of our detectors.

## Figures and Tables

**Figure 1 sensors-23-08061-f001:**
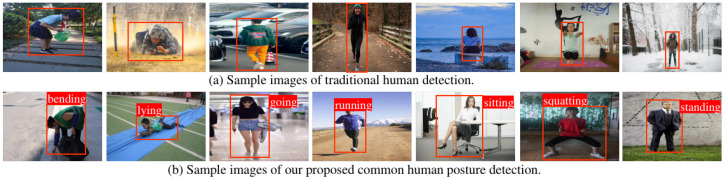
The previous method of human detection primarily concentrated on the identification and localization of humans, as shown in (**a**). In contrast, our method also incorporates additional information, specifically the common human posture, as shown in (**b**). It is worth noting that in (**b**), the common postures of the human (‘bending’, ‘lying’, ‘going’, ‘running’, ‘sitting’, ‘squatting’, and ‘standing’ from left to right) are marked accordingly.

**Figure 2 sensors-23-08061-f002:**
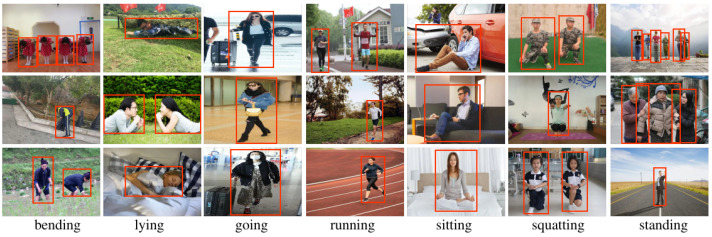
Examples of humans exhibiting seven postures (‘bending’, ‘lying’, ‘going’, ‘running’, ‘sitting’, ‘squatting’, and ‘standing’) are shown in the proposed CHP dataset. The objects are highlighted by red bounding boxes to serve as identifiers.

**Figure 3 sensors-23-08061-f003:**
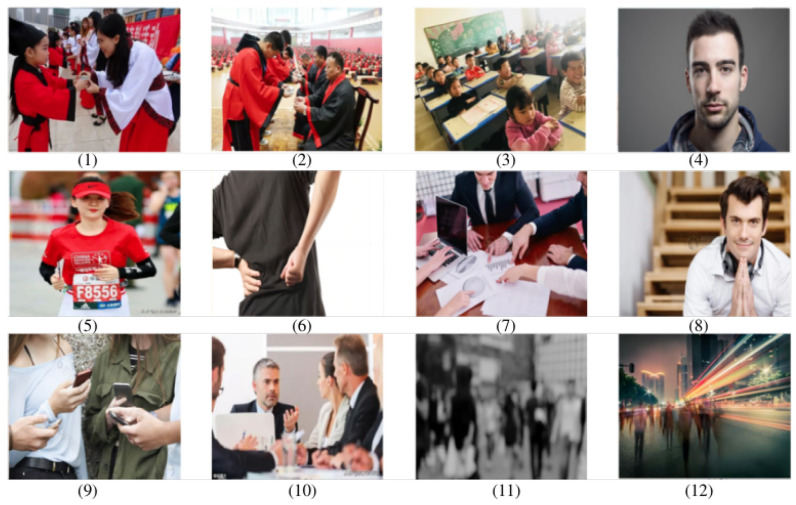
The figure illustrates the types of images that are not suitable for this particular task during the data collection phase.

**Figure 4 sensors-23-08061-f004:**
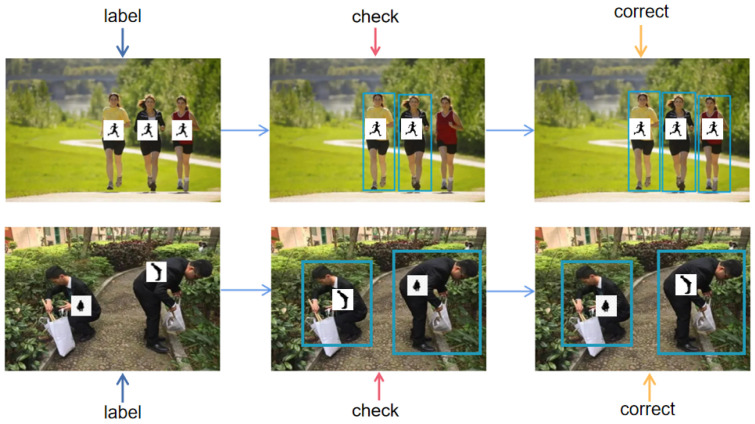
The annotation process for the CHP dataset can be broken down into three key steps: labeling, checking, and correcting. This process involves illustrating how the data are annotated and ensuring accurate annotations are made.

**Figure 5 sensors-23-08061-f005:**
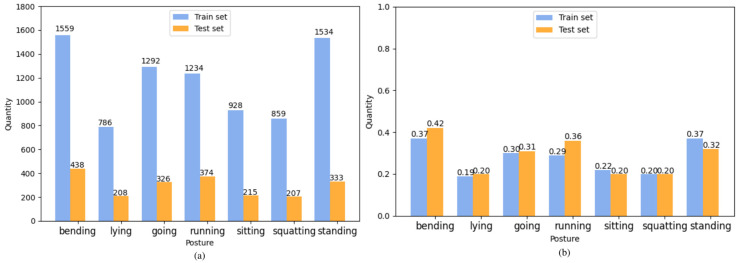
(**a**) shows the number of each posture in in the train set and test set on the CHP, and (**b**) shows the average number of each posture per image in the train set and test set on the CHP.

**Figure 6 sensors-23-08061-f006:**
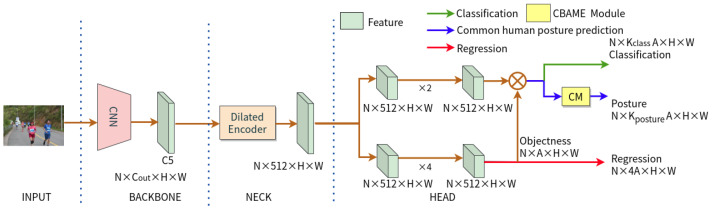
The CHP-YOLOF detector adopts the network structure of IPH-YOLOF, with the sole difference being the integration of the CBAME module into its common human posture prediction head.

**Figure 7 sensors-23-08061-f007:**
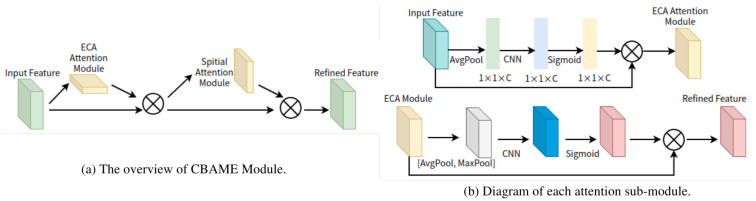
The CBAME module’s network architecture comprises two sequential sub-modules: channel and spatial (from the above figure to the following figure in (**b**)). The channel sub-module utilizes the ECA, which is the difference from the original CBAM. Our module (CBAME) adaptively refines the intermediate feature map at each convolutional block of deep networks.

**Figure 8 sensors-23-08061-f008:**
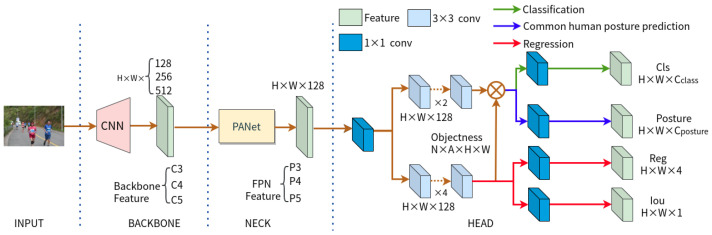
The CHP-YOLOX detector adopts the network structure of IPH-YOLOX, with the only distinction being the utilization of InternImage as its backbone, a CNN foundation model.

**Figure 9 sensors-23-08061-f009:**
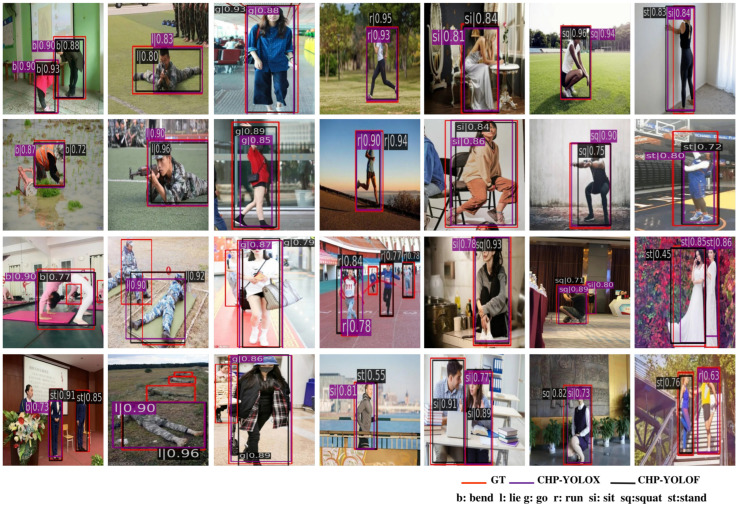
A qualitative evaluation of 28 CHP samples. The last two rows show occasions when detectors predicted incorrectly, while the top two rows show situations where two detectors predicted correctly. The common human postures are indicated by the letter that comes before the vertical line, where the letters b, l, g, r, si, sq, and st stand for, respectively, bending, lying, going, running, sitting, squatting, and standing. GT stands for ground truth.

**Figure 10 sensors-23-08061-f010:**
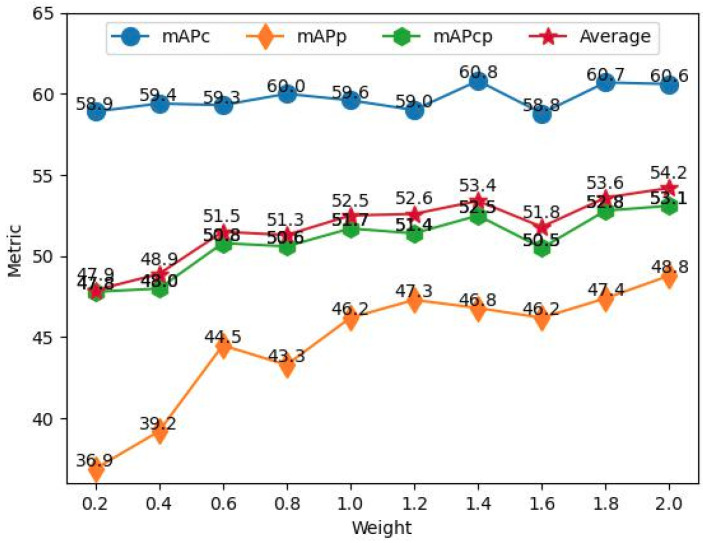
The adjustment of the weighting coefficient for the loss predicting common human posture on the CHP dataset has an impact on the values of the three indicators.

**Table 1 sensors-23-08061-t001:** The $AP_{c}$s of the four original detectors, i.e., YOLOv3, YOLOV5, YOLOF, and YOLOX, are evaluated on the CHP dataset.

Model	{AP${}_{c}$@0.5, AP${}_{c}$@0.75, mAP${}_{c}$}	Model	{AP${}_{c}$@0.5, AP${}_{c}$@0.75, mAP${}_{c}$}
YOLOv3	(0.766, 0.595, 0.520)	YOLOF	(0.868, 0.718, **0.622**)
YOLOv5	(0.825, 0.669, 0.573)	YOLOX	(**0.874, 0.723**, 0.616)

**Table 2 sensors-23-08061-t002:** The APs of the two baseline detectors, namely CHP-YOLOF and CHP-YOLOX, are evaluated on the CHP dataset. It is important to note that $AP_{c}$, $AP_{p}$, and $AP_{cp}$ are the accuracy metrics employed to predict human, common human posture, and combinations of both.

Model	{AP${}_{c}$, AP${}_{p}$, AP${}_{\mathrm{cp}}$}@0.5	{AP${}_{c}$75, AP${}_{p}$75, AP${}_{\mathrm{cp}}$}@0.75	{mAP${}_{c}$, mAP${}_{p}$, mAP${}_{\mathrm{cp}}$}
CHP-YOLOF	(0.818, 0.652, 0.708)	(**0.720, 0.579, 0.632**)	(**0.608**, 0.488, **0.531**)
CHP-YOLOX	(**0.828, 0.696, 0.727**)	(0.692, **0.579**, 0.612)	(0.588, **0.490**, 0.516)

**Table 3 sensors-23-08061-t003:** The mAP${}_{cp}$ for predicting the combination of humans and their postures was compared between the two baseline detectors, i.e., CHP-YOLOF and CHP-YOLOX, on the CHP dataset.

	Bending	Lying	Going	Running	Sitting	Squatting	Standing
(mAP${}_{cp}$)CHP-YOLOF	0.482	0.429	**0.559**	0.546	**0.545**	0.490	0.363
(mAP${}_{cp}$)CHP-YOLOX	**0.498**	**0.452**	0.536	**0.572**	0.511	**0.493**	**0.389**

**Table 4 sensors-23-08061-t004:** Illustrating how the AP metrics of baseline detectors change when different proposed components are applied on the CHP dataset. EM, CM, CT, and IE represent the ECA Module, CBAME Module, Coat, and InternImage, respectively. The symbol × indicates the usage of this component in the corresponding model, while the symbol √ indicates its non-usage.

Method	EM	CM	CT	IE	$\{{\mathrm{AP}}_{c},{\mathrm{AP}}_{t},{\mathrm{AP}}_{\mathrm{ct}}\}@0.5$	$\{{\mathrm{AP}}_{c},{\mathrm{AP}}_{t},{\mathrm{AP}}_{\mathrm{ct}}\}@0.75$	$\left\{{\mathrm{mAP}}_{c},{\mathrm{mAP}}_{t},{\mathrm{mAP}}_{\mathrm{ct}}\right\}$
CHP-YOLOF	×	×	×	×	(0.790, 0.586, 0.659)	(0.704, 0.528, 0.596)	(0.590, 0.441, 0.499)
CHP-YOLOF	√	×	×	×	(0.809, 0.636, 0.694)	(0.717, 0.574, 0.626)	(0.604, 0.481, 0.525)
CHP-YOLOF	×	√	×	×	(0.818, 0.652, 0.708)	(**0.720**, 0.579, **0.632**)	(**0.606**, 0.488, **0.531**)
CHP-YOLOX	×	×	×	×	(0.817, 0.637, 0.677)	(0.673, 0.530, 0.561)	(0.577, 0.449, 0.478)
CHP-YOLOX	×	×	√	×	(0.817, 0.659, 0.698)	(0.652, 0.553, 0.567)	(0.572, 0.459, 0.490)
CHP-YOLOX	×	×	×	√	(**0.828, 0.696, 0.727**)	(0.692, **0.579**, 0.612)	(0.588, **0.490**, 0.516)

**Table 5 sensors-23-08061-t005:** A demonstration of the changes in the AP metrics of CHP-YOLOF in relation to the weighting coefficient for predicting common human posture on the CHP dataset is illustrated.

Model	λ	$\{{\mathrm{AP}}_{c},{\mathrm{AP}}_{p},{\mathrm{AP}}_{\mathrm{cp}}\}@0.5$	$\{{\mathrm{AP}}_{c},{\mathrm{AP}}_{p},{\mathrm{AP}}_{\mathrm{cp}}\}@0.75$	$\left\{{\mathrm{mAP}}_{c},{\mathrm{mAP}}_{p},{\mathrm{mAP}}_{\mathrm{cp}}\right\}$
	0.2	(0.790, 0.499, 0.627)	(0.703, 0.440, 0.576)	(0.589, 0.369, 0.478)
	0.4	(0.790, 0.521, 0.623)	(0.703, 0.470, 0.571)	(0.594, 0.392, 0.480)
	0.6	(0.790, 0.586, 0.673)	(0.704, 0.526, 0.604)	(0.593, 0.442, 0.508)
	0.8	(0.800, 0.570, 0.665)	(0.713, 0.518, 0.605)	(0.600, 0.433, 0.506)
CHP-YOLOF	1.0	(0.800, 0.609, 0.682)	(0.703, 0.554, 0.615)	(0.596, 0.462, 0.517)
	1.2	(0.790, 0.624, 0.676)	(0.704, 0.566, 0.615)	(0.590, 0.473, 0.514)
	1.4	(**0.819**, 0.626, 0.699)	(**0.720**, 0.559, 0.624)	(**0.608**, 0.468, 0.525)
	1.6	(0.799, 0.620, 0.678)	(0.700, **0.601**, 0.603)	(0.588, 0.462, 0.505)
	1.8	(0.818, 0.632, 0.701)	(**0.720**, 0.568, **0.633**)	(0.607, 0.474, 0.528)
	2.0	(0.818, **0.652**, **0.708**)	(**0.720**, 0.579, 0.632)	(0.606, **0.488**, **0.531**)

## Data Availability

The data that support the findings of this study are available from the corresponding author upon reasonable request.

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
