# Peer review of "Beyond Human Detection: A Benchmark for Detecting Common Human Posture"

_sensors, 2023, doi:10.3390/s23198061_

Round 1
Reviewer 1 Report
The manuscript introduced CHP dataset, the first benchmark dedicated to common human posture detection and developed two baseline detectors to support and stimulate further research on CHP.
In the abstract ,authors give the problem ,the solution ,and the experiment result .Readers can easily understand the innovative points of the article.
In Section 1 and 2 authors introduced the background and analyzed a large amount of related works and provided their own contributions.
In Section 3 and 4 ,authors illustrate the details of detection benchmark for human posture and the baseline detectors for detecting CHP .
Finally.Extensive testing shows that the effectiveness and superiority of the proposed algorithms .
The manuscript is well written and organized. However, I have the following concerns.
1,According to template of MDPI ,references should be numbered in order of appearance .
2.What is the model size of CHP-YOLOF and CHP-YOLOX. Is it greater or smaller then the traditional one ?
3.Since the CHP dataset is designed to go beyond human detection and stimulate more informative and promising research directions,does the author consider making this dataset publicly available on GitHub for researchers to download and study?
4.For better readability, the authors may expand the abbreviations at every first occurrence.For example,YOLO, FPN, R-CNN and etc.
Quite good .
Reviewer 2 Report
This paper conducted a study on identifying common human postures.
Conducting experiments and classification on various human postures is a valuable endeavor. However, due to the lack of comparison with similar previous research, it is challenging to determine whether the research conducted in this study is superior or not.
In the past, artificial intelligence techniques have been employed to classify human actions. YOLO has been in use for a long time. At the very least, the proposed technique by the authors should be demonstrated to be superior to the traditional YOLO approach quantitatively through measurable metrics.
This paper formulate a novel task to identify common human postures, which requires detecting humans and estimating their common posures.=>
"This paper formulates a novel task to identify common human postures, which requires detecting humans and estimating their common poses."
---
The first part of Abstract: Human detection involves locating all instances of human beings present in an image, and it finds applications in diverse fields such as search and rescue, surveillance, and autonomous driving. The rapid progress in computer vision and deep learning technologies has significantly enhanced human detection capabilities. However, for more advanced applications like healthcare, human-computer interaction, and scene comprehension, merely localizing humans is insufficient. These applications necessitate a deeper understanding of human behavior and state, enabling effective and safe interactions with humans and their environment.
---
"There are errors in English in several places. Please correct the errors in English."
Round 2
Reviewer 2 Report
Better than before
Small mistake: psoe-> pose
Check English again.
Author Response
Thank you for your valuable comments. We have checked English again in the paper, and the relevant modifications have been highlighted in the paper.